# Investigating the Regulatory Process, Safety, Efficacy and Product Transparency for Nutraceuticals in the USA, Europe and Australia

**DOI:** 10.3390/foods12020427

**Published:** 2023-01-16

**Authors:** Muralikrishna Gangadharan Komala, Ser Gin Ong, Muhammad Uzair Qadri, Lamees M. Elshafie, Carol A. Pollock, Sonia Saad

**Affiliations:** Renal Medicine, Kolling Institute of Medical Research, University of Sydney, St. Leonards, NSW 2065, Australia

**Keywords:** food-based therapeutics, regulations, serialisation technology

## Abstract

Increased numbers of patients with chronic conditions use nutraceuticals or food-based therapeutics. However, to date, there is no global consensus on the regulatory processes for nutraceuticals. With the increased use, issues of quality and safety have also arisen. This review summarises the current regulations held for nutraceuticals in the USA, European and Australian jurisdictions using regulatory authority sites and databases. The efficacy and safety concerns, product development, gaps in regulation and challenges in ensuring product authenticity are also summarised. The data highlight the complexity that the globalisation of nutraceuticals brings with respect to challenges in regulation and associated claims regarding efficacy and safety. The development of an effective system with integrity is needed to increase vertical collaboration between consumers, healthcare practitioners, and government agencies and the development of international risk assessment criteria and botanical compendia. This will help in greater transparency and improved trust in the process and products. Emerging technologies could play a role in improving systems engineering by information sharing and leveraging the strengths of different countries. In conclusion, nutraceuticals have been poorly regulated leading to spurious claims based on little or no real evidence. This makes it difficult to separate meaningful results from poor data. More stringent regulation and an effective system of integrity are required to ensure efficacy and safety and enable the adequate monitoring and increase consumer and healthcare professionals’ confidence.

## 1. Introduction

Healthcare professionals and authorities overall agree that food-based therapeutics are generally ‘safer’ than chemically derived treatments. Although the evidence for this distinction is poor, consumers around the world have embraced natural healthcare products to improve health and well-being. Food-based therapeutics are becoming well-established in the USA and Europe and are an emerging discipline in Australia. The global market displays significant growth forecasted to reach US$278 billion by 2024 [1]. At least 60% of adults greater than 65 years of age in the United States use supplements [2]. 77% of Americans take at least one supplement and the dietary supplements market is estimated to value at US$35.6 billion, and is projected to expand at a compound annual rate of 6.9% by 2026 [3]. The use of natural therapeutics has also been increasing in Europe and Australia. In 2014, it was estimated that ~18.8% of the population in Europe take at least one botanical supplement [4]. Interestingly, 9 in 10 Europeans have taken food supplements in their lives and nearly all of them (93%) had done so in 2021 to boost their immune system due to COVID-19 [5]. The dietary supplements market size in Europe was valued at US$ 61.8 billion in 2021 and is projected to expand at a compound annual growth rate of 5.8% by 2030 [6]. Similarly, Austrade estimated that the Australian market for food-based therapeutics has a growth rate of 7% year on year [7]. Three-quarters of Australians take at least one dietary supplement and a quarter of the population visit complementary healthcare practitioners each year [8]; 24% of Australian adults with a chronic condition utilise complementary and alternative medicines for the treatment of diabetes, arthritis, osteoporosis, and heart conditions. 

The market is flooded with terms such as functional foods, food supplements, herbal extracts, nutraceuticals, food for medicine, etc. There is no consensus around the nomenclature but in general, nutraceuticals refers to a “food or part of a food that provides medical or health benefits including the prevention and/or treatment of a disease” [9] whereas food supplements refer to single substances used alone or in mixtures to restore micronutrient deficiency [10]. Nutraceuticals do not easily fall into a legal category of food or drugs but inhabit a grey area in between and to date, there is no global consensus on the regulation of nutraceuticals. They often do not fall within existing regulatory frameworks such as the Food and Drug Authority (FDA), the European Medicines Agency (EMA) and the Therapeutic Goods Administration (TGA).

Although regulatory regimens are in place, there is often limited scientific information about the safety of nutraceuticals, their efficacy and their mechanism of action [11]. Often evidence for efficacy is not available or not required due to the perceived ‘low risk’ of natural products and challenges in the identification of high-risk nutraceuticals requiring biological and toxicology data. Because of the expanding use of nutriceuticals use and the increased number of product recalls, aspects of efficacy, quality and safety have arisen that call for enhanced regulation.

This review aims to compare and contrast the three main regulatory frameworks, i.e., the FDA, EMA and TGA for therapeutic food products. In addition, the role of the ancillary regulatory authorities for food products such as the Dietary Supplement Health and Education Act (DSHEA), the European Union Food Law legislation and Food Standards Australia and New Zealand (FSANZ) will also be discussed. The literature search included will shed light on the limitations of existing regulations in terms of assessing claimed efficacy and ensuring consumer safety. It will also propose potential changes to the regulatory framework to increase trust, ensure the safety of consumers, and deliver high-quality products and adequate monitoring.

## 2. Regulation of Nutraceuticals

As stated above, there is no consensus on the definition of nutraceuticals and not only the nomenclature but also the regulation varies between different countries. This is summarised in Table 1.

### 2.1. USA 

In general, nutraceuticals such as pro and pre-biotics and special foods for medical use require clinical evidence evaluated for safety and efficacy compared to food supplements such as vitamins, minerals, protein supplements and herbal products [11]. According to the United States Government Office, Food and Drugs Administration (FDA) and Dietary Supplement Health and Education Acts (DSHEA), food supplements are defined as ‘A product (other than tobacco) in the form of a capsule, powder, softgel or gelcap intended to supplement diet to enhance health. The product must contain one or more of the following dietary ingredients: a vitamin, mineral, amino acid, or other botanical or dietary substance’. Zeisel defined nutraceuticals as ‘a dietary supplement that delivers a concentrated form of a biologically active component of food in a non-food matrix to enhance health in doses that exceed that can be found in normal foods’ [12].

In 1938, a set of laws known as the Federal Food, Drug and Cosmetic Act (FFDCA) were passed which allow the FDA to regulate and govern the safety of cosmetics, drugs and food. The FFDCA categorised products under two main streams: food and drugs. Both streams have different regulatory frameworks. In 1994, The DSHEA was passed, introducing a novel stream of foods known as dietary supplements [13] which means that nutraceuticals are regulated as a food category and must meet all provisions for conventional foods specified in the FFDCA. 

The DSHEA aims to distinguish between food and drugs regulatory frameworks and became the fundamental body that governs the regulation of dietary supplements in the USA. Based on this act, manufacturers are responsible for the efficacy, safety, and quality assurance of their products by illustrating the explicit ingredients used in the product and any potential allergies [13]. The acts aim to establish a clear and uniform definition of the term dietary supplement and outline the beneficial qualities of using dietary supplements and enable the FDA to assert jurisdiction over the establishment of Good Manufacturing Practices (GMPs) that are specific to the dietary supplement industry. The DSHEA promotes scientific studies of dietary supplements and encourages research funding to investigate the health benefits of dietary supplements. Clinical trial data and toxicological data are required for various scenarios based on whether the anticipated exposure to the substance exceeds historical consumption. As a result, the Office of Dietary Supplement Regulation was established within the National Institutes of Health to ensure that the survey requirements set by the DSHEA are met [14]. However, the Office of Dietary Supplement Regulation does not play a role in implementing the regulations outlined by the DSHEA. 

The FDA monitors the marketplace for unsafe practices or misleading advertising claims and labelling and is responsible for evaluating premarketing evidence of safety for novel ingredients. No premarketing authorisation is required except for new ingredients not marketed in the USA before 1994 [14] as historical dietary ingredients in dietary supplements, sold in the USA before the implementation of the DSHEA, are considered safe [15].

### 2.2. Europe

Unlike in the US where the FDA is responsible for the safety and regulation of drugs and food, in Europe, food and drug safety are regulated by distinct agencies [15]. According to FUFOSE–Functional Food Science in Europe “A food can be regarded as functional if it satisfactorily demonstrates beneficial effects on one or more target functions in the body, beyond adequate nutritional effects, thus either improving the general physical conditions or/and decreasing the risk of the generation of diseases” [16,17].

The legal characterisation of nutraceuticals in the EU is, in general, made on the basis of accepted effects on the body. If a product contributes to the maintenance of organs or health tissues, it may be considered a food ingredient; however, if it can modify a physiological process it will likely be considered a medicinal substance [18]. The European Medicines Agency (EMA) is responsible for the scientific evaluation, supervision and safety monitoring of medicines. In 2002, the European Parliament and the Council adopted Regulation (EC) No 178/2002 which frames the general principles and requirements of food law (General Food law regulation). This law is the foundation that lay down general principles, guidelines, requirements and procedures for decision-making matters related to food safety, food production and distribution. It also established the EU Food Safety Authority (EFSA), an independent agency responsible for scientific advice and risk assessment. It also develops procedures to manage emergencies to ensure a high level of health protection.

Nutraceuticals in the EU comprise a broad range of products that can be used for feed materials, feed additives, or medicinal purposes. The EU legislation for the different types of nutraceuticals is very complex as discussed in multiple other reviews [18,19,20,21,22]. Different regulatory procedures exist for each of these categories to obtain marketing authority. For example, food for special medical purposes or nutraceuticals that have a medical claim or a pharmacological effect must comply with Directive 2001/82/EC and are dealt with under Regulation (EU) No 1924/2006 and No 609/2013 to ensure efficacy and safety [19,21]. According to the Scientific Committees and panels of the EFSA, biological and toxicology data are required for foods with medical claims [21]. The complexity in the regulation of nutraceuticals allows marketing authorisation with a high safety margin and precaution necessary for human health. 

### 2.3. Australia

The Therapeutic Goods Administration (TGA) in Australia has published Australian regulatory guidelines for complementary medicines in 2018 (version 7.2) [23]. This document describes that products can be classified as either therapeutic goods or medicines. Therapeutic goods come under the ambit of the TGA, and food products come under the ambit of Food Standards Australia and New Zealand (FSANZ). The TGA’s food-medicine interface guidance tool can be utilised to identify a product as either food or therapeutic good. It is to be noted that the Secretary of the Department of health through the TGA, can determine a particular product to be a therapeutic good even if it has a classification under the FSANZ as a food product. 

Medicinal products containing an active ingredient, for example, vitamins, herbal products, etc., are referred to as complementary medicines and regulated as medicines by the TGA. Complementary medicines can be either listed or registered under the TGA. Complementary medicines which are listed do not undergo an assessment before listing and only require certification by the sponsor prior to marketing. However, complementary medicines which are registered require tighter scrutiny and assessment before they are approved for marketing. These medications are listed on the Australian Register for Therapeutic Goods (ARTG). It is to be noted that most complementary medicines are listed and not registered under the ARTG [23].

The TGA tends to align most of its regulatory approach to therapeutic goods with comparable international regulatory counterparts when possible. These include the European Food Safety Authority (EFSA) for the safety and quality of ingredients used in listed medicines and the Food and Drug Administration (FDA) for the safety, quality and efficacy of the registered complementary medicines and certain listed complementary medications [24].

Australia and New Zealand have established FSANZ, a Trans-Tasman regulatory system that develops food standards for Australia and New Zealand. These regulations include food safety, microbiological limits and processing standards. They also deal with food fortification with vitamins and micronutrients and set guidelines for the same [25]. 

In summary, Australia adopts a unique approach suited for this jurisdiction. There is a separation of therapeutic goods and food products. The TGA deals with complementary medicines and the FSANZ regulates food products. The TGA does tend to identify comparable regulatory bodies based on several criteria to determine that they align with the TGA’s regulatory framework. This is to help in international collaboration and shorten evaluation time frames. Once these comparable regulatory bodies are identified, reports from these bodies on complementary medicines are accepted if these reports satisfy certain criteria. However, most complementary medicines do not undergo significant scrutiny unless they are registered with the ARTG. Hence, there remains a gap in the regulation of complementary medicines.

## 3. Nutraceuticals: Efficacy and Safety Concerns 

### 3.1. Efficacy

Nutraceuticals have been documented to have benefits. Selected examples of documented benefits of some nutraceuticals are described in Table 2. A meta-analysis of 12 studies found cinnamon reduced inflammatory markers and could be used as a potential adjuvant in oxidative stress and inflammation after judicious consideration of cost-benefit and patient preference [26]. A meta-analysis of five RCTs finds that patients who had been given *Ganoderma lucidum* (Lingzhi mushroom) alongside chemotherapy/radiotherapy were more likely to respond positively compared to either chemotherapy or *Ganoderma lucidum* on their own [27]. In all studies, modest benefits were found with no/minor and transient ADRs but caution is advised [27].

### 3.2. Safety

In recent years, the potential benefits of nutraceuticals have been shadowed by safety concerns. There is a growing list of products recalled by regulatory authorities in various jurisdictions (Table 3). The reasons for complementary drug recall include undisclosed active ingredients and incorrect labelling [34]. Similarly, dietary supplements have been recalled for undisclosed ingredients and microbiological contamination. However, nutraceuticals even in recommended doses can lead to toxicities mainly because a safe dose may be unknown due to the lack of longitudinal or controlled clinical studies in humans. Green tea as an example has been noted to cause hepatotoxicity [35]. Consumption of around 12 g of garlic daily can have antiplatelet effects and leads to bleeding abnormalities [36]. Failing to accurately identify and list all the ingredients and the actual dose poses a health risk not only due to potential allergic reactions but also due to interactions with other drugs, e.g., the interaction of St. John’s wort with psychotropic medicines, anticoagulants and concerns about its potential effect on the efficacy of other pharmaceuticals and oral contraceptives [37]. Some types of flavonoids such as soy-derived isoflavones have been shown to cause reproductive tract malformations when taken as purified supplements and a few studies showed developmental toxicity and increased risk of Kawasaki disease in children consuming soy milk or formula [38,39,40,41,42]. Fish oil and omega-3 supplements can exacerbate bleeding, which is more problematic in people taking other anticoagulants [43,44,45]. The recommended dosages for different supplements are available although these are highly variable [46]. Nutraceuticals in pharmacological or higher doses of dietary supplements can have unexpected adverse effects through pro-oxidative or alternate mechanisms [47]. Antioxidant supplements have been shown to promote metastases in animal studies and hence should be used with caution in cancer treatment [48]. Similarly, heavy metal contaminants, pesticide residues and fungal metabolites are other risk factors. Increased intake may not be always benign and can lead to adverse effects due to the lack of knowledge that the required threshold of supplements can vary depending on the person’s health condition, i.e., the vitamin dose required for patients with chronic kidney disease and risk of toxicity [49]. According to the TGA’s Regulation Impact Statement for medicines labelling requirements, current labelling design requirements fall short of international standards in the consistent placement of active ingredients or critical health information [50]. To date, 1.3% (2050/152,691) of ADR reports in the TGA Australian Adverse Drug Reactions (ADRS) database are coded as possible drug interactions with 3% (240/12,000) involving complementary medicines, ~94% prescription medicines and 3% OTC in 2002 [51]. 

Despite the regulatory regimes in the above-mentioned jurisdictions, it is apparent that appropriate or excessive use of nutraceuticals results in adverse events leading to safety alerts and/or food recalls (Table 3). This is often due to microbial, biotoxin, chemicals or foreign matter contamination, undeclared allergens, labelling errors, packaging fault, questionable stability, or incorrect potency [52]. Hence, there is scope for better identification and regulation of potentially high-risk nutraceutical products. There is an even more important role for better education of the public in terms of labelling to promote the appropriate use of these agents. Tighter penalties for breaches can add another layer of safety for consumers. 

**Table 3 foods-12-00427-t003:** Data on food safety alerts /recalls collected from respective authorities: USA- FDA, United States Department of Agriculture (USDA), EU- Rapid Alert System for Food and Feed (RASFF) and AUS- FSANZ, TGA and Product Safety Australia (PSA) from years 1999 to 2021 [34,53,54,55,56].

Year	Jurisdiction and Authority				
	US		EU	AUS	
	FDA Recalls	USDA Recalls	RASFF Alerts	FSANZ Recalls	TGA Safety Alerts
2021	342	63	2634	83	40
2020	485	50	1146	109	69
2019	337	124	1175	87	64
2018	459	125	1118	100	71
2017	9199	131	942	69	107
2016	8305	122	847	72	108
2015	9178	150	775	81	107
2014	8061	94	751	76	128
2013	8044	75	596	42	92
2012	9469	82	547	60	72
2011	9288	103	635	67	50
2010	9361	70	592	53	33
2009	8065	69	578	52 (PSA)	10
2008	5778	54	549	43 (PSA)	17
2007	5585	58	961	48 (PSA)	5
2006	4266	34	934	54 (PSA)	1
2005	5338	53	956	61 (PSA)	5
2004	4670	-	691	61 (PSA)	2
2003	4627	-	454	70 (PSA)	3
2002	5025	-	434	46 (PSA)	7
2001	4563	-	302	43 (PSA)	7
2000	3716	-	133	24 (PSA)	2
1999	3736	-	97	41 (PSA)	1

## 4. Proposed Changes to the Regulatory Framework to Ensure Consumers Safety, High-Quality Products and Adequate Monitoring

The EFSA must approve or authorise in detail any health claims before the products are registered in Europe. Subsequently, the member states may require further authorisation. There are no such requirements before product registration in the US. However, the EFSA does not have the authority to investigate nutraceuticals once they are registered, whereas the FDA reserves the right to act against any unsafe product in the market [11]. 

This brings us to the next issue, the harmonisation of regulatory guidelines. The definition of nutraceuticals and the regulatory authority over them are variable in different jurisdictions. For example, the European regulations, define food categories and include a definition of food supplements, it does not recognise the term nutraceuticals [57]. In the USA, the FDA recognises the term nutraceuticals and applies a different set of rules for them as compared to conventional foods and drugs. Hence, legislative and regulatory consistencies will go a long way in preventing duplication of efforts and a better understanding of different products across different regimes and improve consumer safety.

The process of evaluation of nutraceuticals has the potential for improvement. The first step is a scientific hypothesis with a particular health target which must be tested through clinical trials. The scientific data are then assessed with the data on potential benefits and adverse effects before being approved [11]. This can be labour and time intense although this will provide the safest option for consumers. We propose a framework for collective transparent database development and nutraceutical development protocol (Figure 1) adapted with changes from a proposal by Santini et al. [11]. Our proposal includes a safe development protocol with post-marketing surveillance and feedback to regulatory authorities for quality control and analysis of efficacy and adverse outcomes. This continuous feedback loop safeguards the safety of consumers and helps in the development of high-quality nutraceuticals and analogs.

### Product Transparency: Serialisation and Blockchain Integration

There exists a trust gap between consumers and nutraceutical manufacturers. This is primarily due to the authenticity of the source and the ingredients listed in the nutraceuticals. The second reason is inadequate evidence for the supposed efficacy of the products. The second issue can only be regulated by government intervention with stricter criteria and registration. However, there are ways to ensure the authenticity of nutraceuticals and improve transparency. This can be attained by serialisation and blockchain integration. Pharmaceutical serialisation regulations exist today in many nations [58]. In an effort to combat counterfeit drugs, the EU and US have implemented legislation to mandate serialisation. In the EU, The Commission Delegated Regulation 2016/161 under Falsified Medicines Directive (Directive 2011/62/EU) details rules concerning serialisation with unique identifiers and anti-tampering device verification features that must appear on medicines packaging from February 2019 [58]. In the US, the Drug Supply Chain Security Act 2017 mandates unique serialisation numbers and bar codes to build an electronic identification system to trace prescription drugs through distribution. Although the serialisation approach in the US (‘transaction model’) differs from Europe (‘authentication model’), the legislation in both jurisdictions is a major step in providing clear requirements and guidance against counterfeits [58]. This process of serialisation can be applied to nutraceuticals to protect against counterfeit products.

The global nutraceutical packaging market for functional food, beverage and pharmaceutical products considering product form, suitable materials and requirements is projected at US$ 3 Bn in 2019, and forecasted to surpass US$ 5 Bn by 2029 [59]. Blockchain is a new artificial intelligence technology that has seen increased adoption in various sectors over the last 2 years from banking, and logistics (shipping) to the food supply chain and has the potential to be an effective integrative solution. Blockchain provides a neural network of shared, unalterable ledgers for recording the history of transactions by a network of volunteer computers instead of a single institution, making it much harder to alter data after the event. Blockchain technology can effectively handle data integrity and confidentiality, enhance security and ensure transparency. Integration of blockchain technology with deep learning, which is now possible, can provide additional benefits. Not only can it improve data market management, data traceability and security but it can also predict side effects and disease development and automatically recommend authorities’ requests to recall expired or potentially harmful products. The principles of blockchain technology and the benefits of integrating this technique with deep learning were recently reviewed in detail by Shafay et al. [60].

French retailer Carrefour SA has seen a boost in sales on the Blockchain system in 2018 to track 20 items including meat, milk and fruit with IBM (International Business Machines Corporation, an American multinational technology company) and will add 100 more items this year with baby and organic products [61]. Walmart and IBM are currently collaborating with 10 other companies to test scale 25 different items on Blockchain after a successful 2-year pilot to track and locate Mexican mangoes and Chinese pork [62]. Australia’s natural health company Blackmores and Australia Post and two New Zealand-based companies’ dairy exporters Fonterra and New Zealand Post established the Food Trust Framework based on Blockchain in May 2018 [63]. Entry into the blockchain network varies in cost and process sophistication where companies can work with existing systems to build an application programming interface (API) [62]. Suppliers can upload their product’s data using smart devices and capture information on-site more effectively with unique identification codes while consumers upload their user experience or ADRs via an internet portal. Hence, a combination of different strategies can be employed to provide authentic products to the consumer and thereby improving consumer confidence in the system and the product.

## 5. Conclusions

Nutraceuticals are differentially regulated in the USA, Europe and Australia. Gaps in regulation, challenges in technical analysis and global internet sales easing cross-border purchases have facilitated the increase in significant public health risks and economic burden. Nutraceuticals have demonstrated potential benefits but the development of a holistic system with integrity is needed to harness its full potential. Enhanced education to increase awareness and discretion in consumers and healthcare practitioners, careful adherence to good manufacturing practices by industry members, monitoring by national authorities, protocol development, harmonisation of international regulatory regimes and risk assessment criteria are needed. Product traceability can offer transparency and recall efficiency. Resource limitations and quality of data entry can undermine system effectiveness and do not guarantee adherence nor prevent fraudulent activities. The nutraceutical industry is still in its infancy in Australia and the creation of an effective national/international regulatory system with the adoption of Blockchain technology can provide a safer option for consumers and help the nutraceutical industry develop a safety profile and brand value.

## Figures and Tables

**Figure 1 foods-12-00427-f001:**
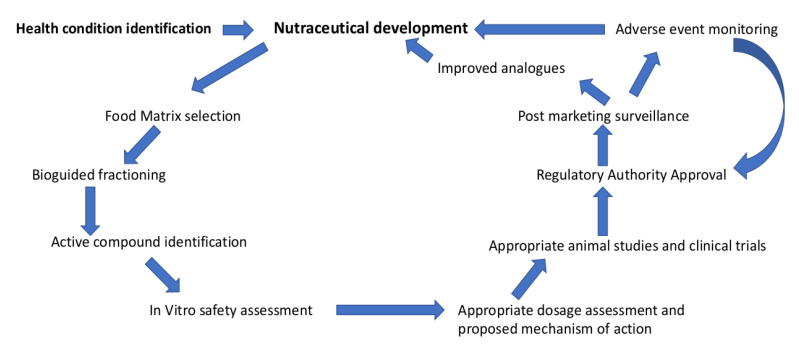
Proposed protocol for development of new nutraceuticals.

**Table 1 foods-12-00427-t001:** The definition of nutraceuticals and their regulation varies between different countries.

	United States	Europe	Australia
**Products Characterisation**	Special foods for medical use or food supplements	Food ingredient or medicinal substance depending on the effect on the body	Food or therapeutic goods according to the TGA interface guidance tool
		Comprise a broad range of products that can be used for feed materials, feed additives or medicinal purposes	Medicinal products containing an active ingredient (referred to as complementary medicines) can be listed or registered
**Regulation Requirement**	Premarketing authorisation is not required except for new ingredients not marketed in the USA before 1994	Different legislation for different types of nutraceuticals.	Most complementary medicines do not undergo significant scrutiny unless they are registered with the ARTG
	Clinical trial data and toxicological data are required if anticipated use exceeds historical consumption.	Biological and toxicology data are required for products with medical claims.	Complementary medicines which are listed only require certification by the sponsor prior to marketing
**Responsibility**	The FDA monitors the marketplace for unsafe practices or misleading advertising claims and labelling and is responsible for the regulation and for evaluating premarketing evidence of safety for novel ingredients	Regulated by the EFSA, an independent agency responsible for scientific advice, risk assessment and for developing procedures for emergencies to ensure safety	TGA deals with complementary medicines, but the FSANZ regulates food products.

**Table 2 foods-12-00427-t002:** Examples of nutraceuticals with therapeutic effects reviewed by Cochrane Systematic Reviews between the years 2004 and 2022.

Cochrane Systematic Review	Study Type	Outcomes	Conclusion
Cinnamon [26]	Systematic review and meta-analysis of 12 studies to elucidate the impact on inflammation and oxidative stress. Daily quantity taken for cinnamon was about 1.5 g–4 g	Primary outcomes were different in different studies and included C-reactive protein (CRP), Malondialdehyde (MDA), total antioxidant capacity (TAC) and intracellular adhesion molecule 1 (ICAM-1). Cinnamon reduced CRP, MDA and TAC with no change in ICAM-1 levels.	Cinnamon supplement can be a potential adjuvant whilst dealing with oxidative stress and inflammation
Lingzhi [27]	Meta-analysis of 5 RCTs comparing the efficacy of *Ganoderma lucidum* with placebo or cancer treatments in patients diagnosed with cancer. All types of cases and stages of cancer were included	Primary outcomes were tumour response evaluated according to the World Health Organization (WHO) criteria, quality of life measured by Karnofsky scale score, natural killer (NK) cell activity and T-Lymphocyte co-receptor subsets. In this meta-analysis, the patients treated with *Ganoderma lucidum* alongside chemo/radiotherapy were shown to have better outcomes compared to each agent on their own.	Trial quality was suboptimal. The meta-analyses show some modest benefits when *Ganoderma lucidum* was administered together with standard chemotherapy. Judicious use after thorough consideration of cost-benefit and patient preference is suggested.
Omega-3 Fatty Acids [28]	Meta-analysis of 86 RCTs involving 162,796 participants comparing intake of high versus low omega-3 fatty acids for at least a year on heart and circulatory disease	Primary outcomes were coronary events and death due to coronary artery disease and all-cause mortality. In this meta-analysis, omega-3 fatty acids reduced the incidence of coronary events and death due to coronary artery disease with no effect on all-cause mortality.	Omega-3 supplementation may reduce coronary events and death due to coronary artery disease.
Probiotics [29]	Meta-analysis of 31 RCTs with 8672 participants determining outcomes of *Clostridium difficle* associated diarrhea (CDAD) in adults and children receiving probiotics	The primary outcome was the incidence of CDAD. These meta-analyses suggested that probiotics in association with antibiotics are effective in preventing CDAD.	Short-term use of probiotics in association with antibiotics is safe and effective in preventing CDAD in hospitalised patients.
Omega-3 Fatty Acids [30]	Meta-analysis of 5 studies with 105 participants determining morbidity and mortality outcomes in patients with cystic fibrosis receiving omega-3 fatty acids	Primary outcomes were the morbidity and mortality outcomes and adverse events in patients with cystic fibrosis treated with omega-3 fatty acids. There was no mortality in the study and moderate morbidity benefits among patients receiving the intervention.	Omega-3 fatty acids may be beneficial in reducing morbidity associated with cystic fibrosis with no significant adverse effects.
Dietary fibre [31]	Meta-analysis of 23 RCTs with 1513 participants determining outcome of dietary fibre on cardiovascular disease and related risk factors	Primary outcome was the development of cardiovascular disease and the modification of risk factors. None of the studies reported on cardiovascular events. There was a significant improvement in total cholesterol levels, LDL cholesterol and diastolic blood pressure.	Dietary fibre can be used as a supplement to improve cardiovascular risk factors
Antioxidant vitamin and/or mineral [32]	Meta-analysis of 19 RCTs to determine outcome of antioxidant vitamin and/or mineral supplementation in preventing progression of age related macular degeneration (AMD)	Patients with AMD may observe some delay in progression with this intervention. The evidence was of low certainty and caution has been suggested while prescribing vitamin supplementation	Vitamin and/or mineral supplementation may be useful in slowing the progression of AMD. However, monitoring for adverse effects of supplementation is essential.
Iron [33]	Summary of evidence from 75 systematic reviews to determine if dietary iron supplementation prevents and controls anaemia in the healthy population	Daily iron supplementation improved the haemoglobin level and reduced the risk of anaemia in infants, preschool and school-aged children and pregnant and non-pregnant women	Dietary iron supplementation may help reduce the risk of iron deficiency anaemia.

## Data Availability

Not applicable.

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
