# Peer review of "Investigating the Regulatory Process, Safety, Efficacy and Product Transparency for Nutraceuticals in the USA, Europe and Australia"

_foods, 2023, doi:10.3390/foods12020427_

Round 1

Reviewer 1 Report

Thank you for the opportunity to review the paper. 

(1) The aim of the research is not explicitly described. The sentence in abstract is described “The development of an effective system with integrity is needed to increase vertical collaboration between consumer, healthcare practitioners, government agencies and the development of international risk assessment criteria and botanical compendia.” I am not sure what issue is being addressed by developing an effective system. I assume that it is trust issue between consumer, healthcare practitioners, government agencies - can you clarify. 

(2) In the introduction, it says “Food based pharmaceuticals are becoming well established in the USA and Europe and is an emerging discipline in Australia.” -However, the data is about Australia market. Please add some data of US and European markets to make a comparison.

(3) Apart from regulation by government agencies, does other monitoring by third-party organizations, media, consumers, etc. have an impact on the safety, efficacy and product transparency for nutraceuticals?

(4) Overall the paper lacks a clear structure. It needs extensive rewriting to link the regulatory process, safety, efficacy and product transparency for nutraceuticals to the issues to be addressed. In addition, specific examples could be added at the start of the introduction to illustrate issue with the regulatory process, safety, efficacy and product transparency for nutraceuticals.

(5) Section 2 provides an introduction to the three regulatory frameworks, could a table be made to illustrate the similarities and differences between FDA, EMA and TGA for therapeutic food products.

(6) What is the purpose of the titled of section 4 - Potential Changes to the the regulatory framework, nutraceutical development and improved product transparency? Why is it “potential changes”?

(7) More space may be needed to elaborate on how emerging technologies work.

(8) Minor suggestion: Table 1 should be spruced up to ensure that the text does not 

Author Response

Thank you for the opportunity to review the paper and for the comments. Kindly see the following responses to your questions. We hope that the additional info we provided is satisfactory and the manuscript is now appropriate for publication in Foods Journal. 

(1) The aim of the research is not explicitly described. The sentence in abstract is described “The development of an effective system with integrity is needed to increase vertical collaboration between consumer, healthcare practitioners, government agencies and the development of international risk assessment criteria and botanical compendia.” I am not sure what issue is being addressed by developing an effective system. I assume that it is trust issue between consumer, healthcare practitioners, government agencies - can you clarify. 

Answer: The following statement is now added to the abstract to clarify the importance of developing an effective system.

This will help in greater transparency and improved trust in the process and products”.

 (2) In the introduction, it says “Food based pharmaceuticals are becoming well established in the USA and Europe and is an emerging discipline in Australia.” -However, the data is about Australia market. Please add some data of US and European markets to make a comparison.

 Answer: We have added the following data on the US and European markets and provided additional references. “At least 60% of adults greater than 65 years of age in the United States use supplements [2]. 77% of Americans take at least one supplement and the dietary supplements market is estimated to value US$35.6 billion, projected to expand at a compound annual rate of 6.9% by 2026 [3]. The use of natural therapeutics has also been increasing in Europe and Australia. In 2014, it was estimated that~ 18.8% of the population in Europe take at least one botanical supplements [4]. Interestingly, 9 in 10 of Europeans have taken food supplements in their lives and nearly all of them (93%) had done so in 2021 to boost their immune system due to COVID-19 [5]. The dietary supplements market size in Europe was valued at US$ 61.8 billion in 2021 and is projected to expand at a compound annual growth rate of 5.8% by 2030“.

(3) Apart from regulation by government agencies, does other monitoring by third-party organizations, media, consumers, etc. have an impact on the safety, efficacy and product transparency for nutraceuticals?

Answer: Yes, monitoring by multiple parties improves surveillance and improves safety, efficacy and product transparency. Our article focusses on the regulatory and legislative aspects of monitoring and the other aspects are not within the scope of this paper. 

(4) Overall the paper lacks a clear structure. It needs extensive rewriting to link the regulatory process, safety, efficacy and product transparency for nutraceuticals to the issues to be addressed. In addition, specific examples could be added at the start of the introduction to illustrate issue with the regulatory process, safety, efficacy and product transparency for nutraceuticals.

Answer: We edited the introduction and provided additional examples about the product’s safety and reasons for food recalls. We also summarised the regulatory processes for natural products in a table and explain the advantage of using tracing technologies to improve products traceability, data management and consumers trust and safety.

(5) Section 2 provides an introduction to the three regulatory frameworks, could a table be made to illustrate the similarities and differences between FDA, EMA and TGA for therapeutic food products.

Answer: A new Table is provided (Table 1) which describes how nutraceuticals are characterised in the United States, Europe and Australia. It also describes the requirement for regulation and the responsible body.

(6) What is the purpose of the titled of section 4 - Potential Changes to the regulatory framework, nutraceutical development and improved product transparency? Why is it “potential changes”?

Answer: In this section we proposed changes to the regulatory framework in order to ensure consumers safety, development of high-quality nutraceuticals and adequate monitoring. The title is now changed to reflect that. It now reads: “Proposed changes to the regulatory framework to ensure consumers safety, high-quality products and adequate monitoring”.

(7) More space may be needed to elaborate on how emerging technologies work.

Answer: Additional information about emerging technologies is now added (As follows) and a detailed reference is provided which describes how blockchain technology work and the benefit of integrating this technique to deep learning

Additional section reads: “Blockchain technology can effectively handle data integrity and confidentiality, enhance security and ensure transparency. Integration of blockchain technology with deep learning, which is now possible, and can provide additional benefits. Not only, it can improve data market management, data traceability and security but it can also predict side effects and disease development and automatically recommend authorities requests to recall expired or potentially harmful products. The principles of blockchain technology and the benefits of integrating this technique with deep learning were recently reviewed in details by Shafay et al. [65].”.

(8) Minor suggestion: Table 1 should be spruced up to ensure that the text does not overlap

Answer: It is difficult to fix the table while using the journal template. We have submitted the table (in an excel format) to be printed in landscape orientation.

Reviewer 2 Report

The study investigates the regulatory process of the nutraceuticals in the USA, Europe and Australia. The topic is very interesting and important from a public point of view. In my opinion, the study includes many sections which are important for nutraceuticals. On the other hand, each section could be a little more detailed to give a comprehensive opinion to the reader. My suggestions are given below.

P2, L48-49: Please add space before [5] and [6]

P3, L112: Please add space before [10]

P3, L144: Please add space before [16]

P4, L172: Please write in uppercase Authority

P4, L174: Please add space before [19]

P4, L198 : Please write the Latin name in italics. In addition, do not abbreviate the Genus name please, since it is the first place in the text, Ganoderma lucidum

P4, L200 : Please write the Latin name in italics here and also, at some other parts throughout yje text.

P4, L204: Please add space before [21]

P5, Table 1: Please delete data written three times in the probiotics part

P5, Table 1 : Please write the Latin names in italics

P6, L218-219: In that part, it is written that nutraceuticals even in recommended doses can lead to toxicity. Two examples are given flavonoids and green tea. I agree with that part, since dose is important to say a compound safe or harmful. However, before saying safe or harmful, dose should be discussed. At which doses or for which population, they showed those adverse effects. In addition, only one reference is given for each example and the references are from 2009 and 2012. For toxicological opinions, reports of the safety authorities should be checked and supported by recent papers. Because if the compounds are really unsafe, there should be some other recent studies. In my opinion, more different compounds and their doses should be given as example and should be supported with recent studies and reports, if possible.

Author Response

We thank the reviewer for the comments. We appreciate the opportunity to improve the review. Kindly see below the response to your comments/questions. We hope that the additional info we provided is satisfactory and the manuscript is now appropriate for publication in Foods Journal.

Comments/Questions:

1) The study investigates the regulatory process of the nutraceuticals in the USA, Europe and Australia. The topic is very interesting and important from a public point of view. In my opinion, the study includes many sections which are important for nutraceuticals. On the other hand, each section could be a little more detailed to give a comprehensive opinion to the reader. My suggestions are given below.

P2, L48-49: Please add space before [5] and [6]

P3, L112: Please add space before [10]

P3, L144: Please add space before [16]

P4, L172: Please write in uppercase Authority

P4, L174: Please add space before [19]

P4, L198 : Please write the Latin name in italics. In addition, do not abbreviate the Genus name please, since it is the first place in the text, Ganoderma lucidum

P4, L200 : Please write the Latin name in italics here and also, at some other parts throughout yje text.

P4, L204: Please add space before [21]

P5, Table 1: Please delete data written three times in the probiotics part

P5, Table 1 : Please write the Latin names in italics

Answer: All the above have been rectified.

2) P6, L218-219: In that part, it is written that nutraceuticals even in recommended doses can lead to toxicity. Two examples are given flavonoids and green tea. I agree with that part, since dose is important to say a compound safe or harmful. However, before saying safe or harmful, dose should be discussed. At which doses or for which population, they showed those adverse effects. In addition, only one reference is given for each example and the references are from 2009 and 2012. For toxicological opinions, reports of the safety authorities should be checked and supported by recent papers. Because if the compounds are really unsafe, there should be some other recent studies. In my opinion, more different compounds and their doses should be given as example and should be supported with recent studies and reports, if possible.

Answer: We agree that the references are old. We have edited this section and included additional examples and 10 new references. Doses are variable as noted in the text and toxicity is possible at higher doses. We have attached a recent paper (reference 48) which lists recommended doses for different types of nutraceuticals but often the safe dose is unknown due to the lack of longitudinal or clinical studies.

Section 3.2 is now updated and it reads “Nutraceuticals even in recommended doses can lead to toxicities mainly because a safe dose may be unknown due to the lack of longitudinal or controlled clinical studies in human. Green tea as example has been noted to cause hepatotoxicity [37]. Consumption of around 12 grams of garlic daily can have antiplatelet effects and leads to bleeding abnormalities [38]. Failing to accurately identify and list all the ingredients and the actual dose poses a health risk not only due to potential allergic reactions but also due to interactions with other drugs e.g. the interaction of St. John’s wort with psychotropic medicines, anticoagulants and concerns about its potential effect on the efficacy of other pharmaceuticals and oral contraceptives [39]. Some types of flavanoids like soy derived isoflavones have been shown to cause reproductive tract malformations when taken as purified supplements and few studies showed developmental toxicity and increased risk of Kawasaki disease in children consuming soy milk or formula [40-44]. Fish oil and omega-3 supplements can exacerbate bleeding, which is more problematic in people taking other anticoagulants [45-47]. The recommended dosages for different supplements are available although these are highly variable [48].

Round 2

Reviewer 1 Report

Authors have revised the manuscript well on taking into account my comments, and I agree to accept this work to be published. But authors should check the format of the manuscript to meet the Journal's format requirement. For example, the font in Table 1 is inconsistent with the text.